# Waveform Design for Multi-Target Detection Based on Two-Stage Information Criterion

**DOI:** 10.3390/e24081075

**Published:** 2022-08-03

**Authors:** Yu Xiao, Xiaoxiang Hu

**Affiliations:** 1School of Automation, Northwestern Polytechnical University, Xi’an 710072, China; xxhu@nwpu.edu.cn; 2Air and Missile Defense College, Air Force Engineering University, Xi’an 710051, China

**Keywords:** multi-target detection, MI, KLD, waveform design, information criterion

## Abstract

Parameter estimation accuracy and average sample number (ASN) reduction are important to improving target detection performance in sequential hypothesis tests. Multiple-input multiple-output (MIMO) radar can balance between parameter estimation accuracy and ASN reduction through waveform diversity. In this study, we propose a waveform design method based on a two-stage information criterion to improve multi-target detection performance. In the first stage, the waveform is designed to estimate the target parameters based on the criterion of single-hypothesis mutual information (MI) maximization under the constraint of the signal-to-noise ratio (SNR). In the second stage, the objective function is designed based on the criterion of MI minimization and Kullback–Leibler divergence (KLD) maximization between multi-hypothesis posterior probabilities, and the waveform is chosen from the waveform library of the first-stage parameter estimation. Furthermore, an adaptive waveform design algorithm framework for multi-target detection is proposed. The simulation results reveal that the waveform design based on the two-stage information criterion can rapidly detect the target direction. In addition, the waveform design based on the criterion of dual-hypothesis MI minimization can improve the parameter estimation performance, whereas the design based on the criterion of dual-hypothesis KLD maximization can improve the target detection performance.

## 1. Introduction

A cognitive radar receives and processes a transmission waveform according to the target environment interactive relationship. It optimally allocates limited resources to improve the performance of the radar system in a complex environment [1,2,3,4]. The architecture of the cognitive radar system was proposed by Haykin et al. [5], and a preferred framework for solving the target detection problem lies in combining cognitive radar with a sequential hypothesis test [6]. Compared with the classical hypothesis test, the sequential hypothesis test can adjust a transmission waveform according to previous observation data, making it possible to achieve the expected detection performance within a shorter time.

During sequential hypothesis tests, multiple-input multiple-output (MIMO) radar can improve target detection performance through waveform design [7]. A waveform design based on a sequential hypothesis test framework was proposed in which the radar updated the probability of the target hypothesis via closed-loop feedback [8]. Based on [8], the sum of the products between the characteristic waveform and the probability of each hypothesis was considered as a multi-hypothesis test waveform to classify multiple targets, and good classification performance was achieved in the frequency domain [9]. Romero et al. [10] designed two transmission waveforms matching the target response based on the SNR and the mutual information (MI) criterion and applied them to target classification in the multi-hypothesis test. However, the waveforms designed above do not have adaptive characteristics.

Another way is the sequential waveform design, it can optimize the next transmission waveform according to the previously received echo data. The commonly used criteria include the Reuven–Messer bound (RMB) criterion [11], the Bayesian Cramér–Rao bound (BCRB) criterion [12], and the minimum mean square error (MMSE) criterion [13]. Considering the environmental uncertainties, sequential detection waveforms were designed based on the BCRB criterion to track moving targets [12]. Based on the radar signal model [11], a sequential waveform optimization was proposed, which estimated the angle parameter based on the MMSE criterion [13]. Compared to the methods of predesigning a waveform library, the sequential waveform could change adaptively to the target’s characteristics [14]. Therefore, our present work focuses on sequential waveform design.

Sequential waveform design can be used to flexibly adjust the average number of observation signal samples or to reduce the average sample number (ASN) during sequential hypothesis tests. In MIMO radar detection, a transmission waveform was designed to reduce the ASN by maximizing the distance between composite hypotheses based on the Kullback–Leibler divergence (KLD) criterion [15]. The optimal waveform could balance the signal-clutter-noise ratio (SCNR) and detection probability in the frequency domain [16]. However, it is difficult to acquire the KLD expression between different hypotheses when some parameters are unknown in a complex hypothesis test problem. Using the generalized likelihood ratio test (GLRT) framework, the target parameters were estimated based on the criterion of maximum likelihood (ML) estimation before the waveform design according to the KLD criterion [6]. However, parameter estimation accuracy affected the performance of ASN, and waveforms designed by the KLD criterion could not easily balance parameter estimation accuracy and ASN reduction [17]. To estimate unknown parameters during sequential hypothesis tests, the waveform design criteria that can be used include MI [18,19,20], MMSE [21,22,23], and BCRB [11,12]. Considering that both MI criteria and KLD criteria are information theory criteria, there is a certain constraint relationship between them [24]. Under the sequential hypothesis test framework, the waveform design has been conducted simultaneously based on information criterion like MI and KLD in a few studies [25,26]. Therefore, this paper attempts to optimize the design waveform by combining the MI criterion for parameter estimation and the KLD criterion for ASN reduction.

The main idea is that a waveform library is designed in the first stage, and a waveform is chosen in the second stage. The MI criterion and KLD criterion have mostly been applied to two-stage waveform optimization. Meanwhile, considering that, under practical conditions, the SNR of a tracked target needs to remain within a specified threshold for the radar system to maintain stable tracking of known targets [27]. The SNR should be taken as a constraint condition to adjust and optimize the transmitted waveform. In [28], the information theoretic criteria-based waveform design under SNR constraint was studied from a frequency domain perspective, and the results revealed that the SNR constraint waveform can take full advantage of transmitted energy to extract target signals at frequency points containing targets with weak clutter. However, the waveform diversity performance under the SNR constraint was not analyzed from the perspective of the time domain [29]. According to the above analysis, an adaptive waveform is designed in this study by taking information theoretic criteria as the criterion based on the sequential hypothesis test framework to optimize the target detection performance. The main innovative points are summarized as follows:

1. The MI criterion-based waveform under SNR constraint is proposed to meet the demand for steady target detection. The theoretical analysis indicates that the optimized waveform matrix matches the target response and can take full advantage of transmitted energy to extract target signals.

2. The progressive relationship between parameter estimation and ASN in multi-target detection is obtained, and a two-stage waveform design is proposed. In the first stage, a waveform library for parameter estimation is designed. In the second stage, a waveform is selected from the waveform library according to the waveform optimization criterion of reducing the ASN.

3. The KLD criterion [26] is improved, and the calculated quantity in waveform optimization is reduced. Next, a waveform optimization method reducing the ASN is proposed by considering the MI and KLD between multiple hypotheses as the criterion.

4. An algorithm framework for adaptive waveform design is formulated according to the waveform design characteristics based on the two-stage information theoretic criteria, with the following logical structure. The sequential decision conditions are calculated according to the target parameters. Next, the transmission waveform and prior probability density function (PDF) of each hypothesis are updated. Subsequently, two-stage transmission waveforms are iteratively designed to further update the PDF of parameter estimation, followed by the closed-loop iteration of target parameter estimation.

5. The two-stage adaptive waveform design is simulated and verified by considering three targets of radar detection as an example. The results reflect that the relationship between the parameter estimation accuracy and ASN reduction can be a trade-off. The waveform design based on the dual-hypothesis MI minimization criterion can improve the parameter estimation performance, and that based on the dual-hypothesis KLD maximization criterion can improve the target detection performance.

The remaining sections of this paper are organized as follows. In Section 2, we establish the multi-hypothesis test model and formulate the waveform design problem. In Section 3, we introduce the constraint relations between parameter estimation and ASN. In Section 4, the method of waveform design based on the two-stage information criterion is presented. In Section 5, the simulation results are presented to verify the effectiveness of the algorithm. Finally, we conclude the paper in Section 6.

## 2. Signal Model and Problem Formulation

### 2.1. Radar Signal Model

The transmission waveform matrix of the *k*-th pulse of a standard MIMO radar system [11] is defined herein as $S_{k}$, which consists of *L* different instantaneous samples of the transmission beam. The *l*-th column of $S_{k}$ represents the *l*-th sample data, and the rows of $S_{k}$ correspond to the signals transmitted by $N_{T}$ transmission units composed of linear array elements. Therefore, $S_{k}\in C^{N_{T}\times L}$. The transmitted signal is then received by $N_{R}$ linear array elements after being reflected by the target and the environment.

The observed signal $X_{k}\in C^{N_{R}\times L}$ corresponding to the *k*-th pulse of the transmitted signal can be expressed as [23]:(1)$$X_{k}=H\left(\Phi_{k}\right)S_{k}+N_{k}$$
where $N_{k}\in C^{N_{R}\times L}$ denotes additive white Gaussian noise (AWGN), and $H\left(\Phi_{k}\right)\in C^{N_{R}\times N_{T}}$ represents the target response matrix, which is a nonlinear function of the target parameter $\Phi_{k}$. With Q targets in the environment, the target response matrix can be expressed as [23]:(2)$$H\left(\Phi_{k}\right)=\overset{Q}{\underset{q=1}{\sum }}H\left(\Phi_{k,q}\right)=\overset{Q}{\underset{q=1}{\sum }}\alpha_{k,q}a_{R}\left(\theta_{k,q}\right)a_{T}^{T}\left(\theta_{k,q}\right)$$
where $\alpha_{k,q}$ and $\theta_{k,q}$ represent the attenuation amplitude and direction angle of target *q*, respectively. Moreover, $\alpha_{k}={\left[\begin{array}{cccc} \alpha_{k,1} & \alpha_{k,2} & \cdots & \alpha_{k,Q} \end{array}\right]}^{T}$ and $\theta_{k}={\left[\begin{array}{cccc} \theta_{k,1} & \theta_{k,2} & \cdots & \theta_{k,Q} \end{array}\right]}^{T}$, where the superscript T represents the standard transpose operator, and $a_{T}\left(\theta_{k,q}\right)\in C^{N_{T}\times 1}$ and $a_{R}\left(\theta_{k,q}\right)\in C^{N_{R}\times 1}$ are the steering vectors of the transmitting array and receiving array, respectively:(3)$$a_{T}\left(\theta_{k,q}\right)={\left[\begin{array}{cccc} 1 & {\left(a_{k,q}\right)}^{1} & \cdots & {\left(a_{k,q}\right)}^{N_{T}-1} \end{array}\right]}^{T}$$
(4)$$a_{R}\left(\theta_{k,q}\right)={\left[\begin{array}{cccc} 1 & {\left(a_{k,q}\right)}^{1} & \cdots & {\left(a_{k,q}\right)}^{N_{R}-1} \end{array}\right]}^{T}$$

Here, $a_{k,q}=\exp\left(-j2\pi \left(d/\lambda \right)\sin\left(\theta_{k,q}\right)\right)$, where $j≜\sqrt{-1}$, *λ* denotes the wavelength of the transmission beam, and both arrays are assumed to adopt the same element spacing *d*. Hence, the response matrix of target *q* can be obtained according to the equation of the target response matrix:(5)$$H\left(\Phi_{k,q}\right)=\alpha_{k,q}\left[\begin{array}{cccc} {\left(a_{k,q}\right)}^{0} & {\left(a_{k,q}\right)}^{1} & \cdots & {\left(a_{k,q}\right)}^{N_{T}-1}\\ {\left(a_{k,q}\right)}^{1} & {\left(a_{k,q}\right)}^{2} & \cdots & {\left(a_{k,q}\right)}^{N_{T}}\\ \vdots & \vdots & \ddots & \vdots \\ {\left(a_{k,q}\right)}^{N_{R}-1} & {\left(a_{k,q}\right)}^{N_{R}} & \cdots & {\left(a_{k,q}\right)}^{N_{T}+N_{R}-2} \end{array}\right]$$

As evident from Equation (5), $H\left(\Phi_{k,q}\right)$ includes $N_{T}+N_{R}-1$ different elements $\alpha_{k,q}{\left(a_{k,q}\right)}^{i}$, where $i=0,1,\ldots ,N_{T}+N_{R}-2$. If the target responds as $h_{k}={\left[\begin{array}{cccc} {\left(h_{k}\right)}^{0} & {\left(h_{k}\right)}^{1} & \cdots & {\left(h_{k}\right)}^{N_{T}+N_{R}-2} \end{array}\right]}^{T}$, where ${\left(h_{k}\right)}^{i}=\sum_{q=1}^{Q}\alpha_{k,q}{\left(a_{k,q}\right)}^{i}$, the corresponding covariance matrix is $R_{k}$, then the received signal model in Equation (1) can be transformed into the following vector model:(6)$$x_{k}={\bar{S}}_{k}h_{k}+n_{k}$$
where $x_{k}=\mathrm{vec}\left(X_{k}\right)$, $n_{k}=\mathrm{vec}\left(N_{k}\right)$, and the diagonal matrix $R_{N_{k}}\in C^{N_{R}L\times N_{R}L}$ is the noise covariance matrix of $n_{k}$. The waveform matrix ${\bar{S}}_{k}\in C^{N_{R}L\times (N_{T}+N_{R}-1)}$ is expressed as ${\bar{S}}_{k}={\left[\begin{array}{cccc} {\bar{S}}_{k,1} & {\bar{S}}_{k,2} & \cdots & {\bar{S}}_{k,L} \end{array}\right]}^{T}$, in which each submatrix ${\bar{S}}_{k,l}\in C^{N_{R}\times (N_{T}+N_{R}-1)}$ is a column vector ${\left[\begin{array}{cccc} s_{1l} & s_{2l} & \cdots & s_{N_{T}l} \end{array}\right]}^{T}$ consisting of $N_{T}$ elements in column l of $S_{k}$. The Hankel matrix formed through transposed transformation, i.e., the vector of row j in ${\bar{S}}_{k,l}$, can be expressed as $\left[\begin{array}{cccc} \begin{array}{ccc} 0_{j-1} & s_{1l} & s_{2l} \end{array} & \cdots & s_{N_{T}l} & 0_{N_{R}-j} \end{array}\right]$, where $0_{j}$ represents a row with *j* zero elements.

### 2.2. Multi-Target Detection

Target detection refers to rapid and accurate identification of the target within the field of view of radars. Binary hypothesis testing can determine the presence of a target in the echo signal, while multi-hypothesis testing can determine the number of targets and estimate the target positions [17]. The interesting area of the radar scene is discretized into M angular grids, namely $\left\{{\bar{\theta }}_{1},\cdots ,{\bar{\theta }}_{M}\right\}$, and then multiple hypotheses testing can be used to achieve the goal of multi-target detection. Assuming at most $Q_{\max}$ targets located on these grids, when detecting the *m*-th angle grid ${\bar{\theta }}_{m}$, the total number of multi-hypotheses for the existence of a target is $T=C_{Q_{\max}}^{1}+C_{Q_{\max}}^{2}+\cdots C_{Q_{\max}}^{Q_{\max}}$, where $C_{a}^{b}$ is the operation of combination, which can be calculated as $\frac{a!}{b!(a-b)!}$, where ! denote factorial. Hypothesis $H_{0}$ denotes the null hypothesis, according to which there is no target in the radar scene, and hypothesis $H_{i}(i\geq 1)$ denotes that there is at least one target in the radar scene. In the case study, at most three targets are located on the angle grid ${\bar{\theta }}_{m}$, and the multi-target detection can be described as the multi-hypothesis test model shown below:(7)$$\left\{\begin{array}{cc} H_{1}:\Phi_{k,1}={\left(\alpha_{k,1}\;\;\theta_{k,1}\right)}^{T} & \mathrm{target}1\mathrm{exists}\\ H_{2}:\Phi_{k,2}={\left(\alpha_{k,2}\;\;\theta_{k,2}\right)}^{T} & \mathrm{target}2\mathrm{exists}\\ H_{3}:\Phi_{k,3}={\left(\alpha_{k,3}\;\;\theta_{k,3}\right)}^{T} & \mathrm{target}3\mathrm{exists}\\ H_{4}:\Phi_{k,4}={\left(\alpha_{k,1}\;\;\alpha_{k,2}\;\;\theta_{k,1}\;\;\theta_{k,2}\right)}^{T} & \mathrm{target}1\mathrm{and}2\mathrm{exist}\\ H_{5}:\Phi_{k,5}={\left(\alpha_{k,1}\;\;\alpha_{k,3}\;\;\theta_{k,1}\;\;\theta_{k,3}\right)}^{T} & \mathrm{target}1\mathrm{and}3\mathrm{exist}\\ H_{6}:\Phi_{k,6}={\left(\alpha_{k,2}\;\;\alpha_{k,3}\;\;\theta_{k,2}\;\;\theta_{k,3}\right)}^{T} & \mathrm{target}2\mathrm{and}3\mathrm{exist}\\ H_{7}:\Phi_{k,7}={\left(\alpha_{k,1}\;\;\alpha_{k,2}\;\;\alpha_{k,3}\;\;\theta_{k,1}\;\;\theta_{k,2}\;\;\theta_{k,3}\right)}^{T} & \mathrm{target}1,2\mathrm{and}3\mathrm{exist} \end{array}\right.$$

Here, $\Phi_{k,i}$ is the target parameter of hypothesis $H_{i}$, and is a vector composed of the target attenuation amplitude and the angle parameter.

Given the target parameter $\Phi_{k,i}$, according to Equations (2) and (5), the target response vector $h_{ik}$ of hypothesis $H_{i}$ can be obtained, and the corresponding covariance matrix is expressed as $R_{ik}=h_{ik}h_{ik}^{H}$, where the superscript H represents the conjugate transpose operator. Let us assume that the target response $h_{ik}$ and noise $n_{k}$ are independent. Hence, the probability distribution function of each hypothesized observation signal $x_{k}$ can be expressed as a multi-hypothesis testing model [25]
(8)$$\left\{\begin{array}{l} H_{0}:x_{k}\sim CN\left(0,R_{N_{k}}\right)\\ H_{1}:x_{k}\sim CN\left({\bar{S}}_{k}h_{1k},{\bar{S}}_{k}R_{1k}{\bar{S}}_{k}^{H}+R_{N_{k}}\right)\\ H_{2}:x_{k}\sim CN\left({\bar{S}}_{k}h_{2k},{\bar{S}}_{k}R_{2k}{\bar{S}}_{k}^{H}+R_{N_{k}}\right)\\ \ldots \\ H_{t}:x_{k}\sim CN\left({\bar{S}}_{k}h_{tk},{\bar{S}}_{k}R_{tk}{\bar{S}}_{k}^{H}+R_{N_{k}}\right) \end{array}\right.$$

When the number of targets and parameters are unknown, it is difficult to judge multiple targets using a binary hypothesis testing model. In this paper, multi-hypotheses are designed according to the target number, and only the unknown target parameters need to be considered. According to the hypothesis test framework [8], the multi-hypothesis test can be conducted by analyzing the sequentially received observation signal $x_{k}$. The design of the transmission waveform ${\bar{S}}_{k}$ can realize the parameter estimation of the target [18], and also reduce the ASN for sequential hypothesis testing [17]. However, it is unknown whether the two can be balanced. Therefore, it is necessary to explore the relationship between the ASN and estimated parameters.

## 3. Constraint Relations between Parameter Estimation and ASN

In multi-hypothesis testing, the decision conditions are presented through the sequential probability ratio test. After the irradiation of the *k*-th pulse signal, the probability ratio between hypotheses $H_{i}$ and $H_{j}$ is expressed as:(9)$$\Lambda_{i,j}^{\left(k\right)}=\frac{P_{i}\left(x_{1}|\Phi_{1,i}\right)P_{i}\left(x_{2}|\Phi_{2,i}\right)\ldots P_{i}\left(x_{k}|\Phi_{k,i}\right)\cdot P_{i}^{\left(0\right)}}{P_{j}\left(x_{1}|\Phi_{1,j}\right)P_{j}\left(x_{2}|\Phi_{2,j}\right)\ldots P_{j}\left(x_{k}|\Phi_{k,j}\right)\cdot P_{j}^{\left(0\right)}}$$
where $P_{i}^{\left(0\right)}$ and $P_{j}^{\left(0\right)}$ is the prior probability of hypothesis $H_{i}$ and $H_{j}$, respectively, and $P_{i}\left(x_{k}|\Phi_{k,i}\right)$ denotes the PDF of hypothesis $H_{i}$ parameterized by $\Phi_{k,i}$ when the signal is received for the *k*-th time. Note that $P_{i}\left(x_{k}|\Phi_{k,i}\right)$ depends on the parameter $\Phi_{k,i}$, which can be estimated by particle filter.

The error probability $\varepsilon_{ij}$ is defined as the probability for hypothesis $H_{i}$ to be mistakenly judged as $H_{j}$, namely:(10)$$\varepsilon_{ij}=P\left(D_{j}|H_{i}\right)\;,\;\mathrm{for}\;\mathrm{all}\;i\neq j$$
where $D_{j}$ denotes that hypothesis $H_{j}$ is accepted. According to the sequential probability ratio test model [8,17], the hypothesis decision condition is:(11)$$\Lambda_{i,j}^{\left(k\right)}>\frac{1-\varepsilon_{ij}}{\varepsilon_{ij}}\;,\;\mathrm{for}\;\mathrm{all}\;i\neq j$$

When the hypothesis decision condition satisfies Equation (11), the correct hypothesis is judged as $H_{i}$, otherwise the hypothesis $H_{i}$ is rejected. Then, the newly received data should be continuously observed, and the decision should be made once again. Owing to the error probability $0<\varepsilon_{ij}<1$, Equation (11) can be converted to:(12)$$\Lambda_{i,j}^{\left(k\right)}<\frac{\varepsilon_{ji}}{1-\varepsilon_{ji}}\;,\;\mathrm{for}\;\mathrm{all}\;i\neq j$$

Combining Equations (11) and (12), Wald’s SPRT decision condition is satisfied [30]. As the sequential probability ratio test is a Bayesian method, the prior probability of the target should be accurately understood to acquire the error probability $\varepsilon_{ij}$. For Equation (11), the error probability $\varepsilon_{ij}$ is generally an assigned value. The target detection performance can be effectively enhanced by adaptive adjustment of $\varepsilon_{ij}$, but the adaptive design of $\varepsilon_{ij}$ is beyond the scope of the current discussion.

To judge multiple hypotheses rapidly, the probability ratio $\Lambda_{i,j}^{\left(k\right)}$ should be discussed. Based on the definition of the PDF, $P_{i}\left(x_{k}|\Phi_{k,i}\right)$ can be denoted by a circularly symmetric complex Gaussian PDF as [17]:(13)$$P_{i}\left(x_{k}|\Phi_{k,i}\right)=\frac{1}{\sqrt{{\left(2\pi \right)}^{LN_{R}}\det\left(R_{N_{k}}\right)}}\times \exp\left[-{\left(x_{k}-{\bar{S}}_{k}h_{ik}\left(\Phi_{k,i}\right)\right)}^{H}R_{N_{k}}^{-1}\left(x_{k}-{\bar{S}}_{k}h_{ik}\left(\Phi_{k,i}\right)\right)\right]$$

In Equation (13), $P_{i}\left(x_{k}|\Phi_{k,i}\right)$ is mainly decided by the transmission waveform ${\bar{S}}_{k}$ and the target parameter $\Phi_{k,i}$. Under a fixed transmission waveform, the precision of the target parameter $\Phi_{k,i}$ plays a decisive role in the decision of the sequential probability ratio test according to Equation (11). The general form [13] for estimating the target parameter $\Phi_{k,i}$ is:(14)$${\overset{⌢}{\Phi }}_{k}=E\left(\Phi_{k}|x_{k},{\bar{S}}_{k},x^{k-1},{\bar{S}}^{k-1}\right)$$

Here, $x^{k-1}$ and ${\bar{S}}^{k-1}$ represent the signal accumulation matrices of the observed signal and the emitted signal in the first k−1 snapshots. As parameter estimation is not the focus of this research study, a particle filter is employed to estimate the target parameters. It is considered that the target parameter PDF can be expressed as [23]:(15)$$\begin{array}{c} P\left(\Phi_{k}|x_{k},{\bar{S}}_{k},x^{k-1},{\bar{S}}^{k-1}\right)\propto P\left(x_{k}|\Phi_{k},{\bar{S}}_{k},x^{k-1},{\bar{S}}^{k-1}\right)P\left(\Phi_{k}|{\bar{S}}_{k},x^{k-1},{\bar{S}}^{k-1}\right)\\ =P\left(x_{k}|\Phi_{k},{\bar{S}}_{k}\right)P\left(\Phi_{k}|x^{k-1},{\bar{S}}^{k-1}\right) \end{array}$$

Here, the definition of the system model in Equation (6) is such that $x_{k}$ has no dependence on $x^{k-1}$, ${\bar{S}}^{k-1}$. Then $P\left(x_{k}|\Phi_{k},{\bar{S}}_{k},x^{k-1},{\bar{S}}^{k-1}\right)=P\left(x_{k}|\Phi_{k},{\bar{S}}_{k}\right)$. In addition, as the probability of the parameter $\Phi_{k}$ does not depend on the transmission waveform ${\bar{S}}_{k}$ unless there are conditions on the observation $x_{k}$, then $P\left(\Phi_{k}|{\bar{S}}_{k},x^{k-1},{\bar{S}}^{k-1}\right)=P\left(\Phi_{k}|x^{k-1},{\bar{S}}^{k-1}\right)$. For a particle filter, any probability distribution $P\left(\Phi_{k}|x^{k-1},{\bar{S}}^{k-1}\right)$ can be approximated with the Monte Carlo approach using a discrete particle set, as shown below [23]:(16)$$P\left(\Phi_{k}|x^{k-1},{\bar{S}}^{k-1}\right)\approx \overset{N_{p}}{\underset{j=1}{\sum }}\omega_{k}^{\left(j\right)}\delta \left(\Phi_{k}-\Phi_{k}^{\left(j\right)}\right)$$
where $\Phi_{k}^{\left(j\right)}$, $\omega_{k}^{\left(j\right)}$, and $N_{p}$ represent the status and weight of the target parameter and the total number of particles, respectively, at *k*-th time. Moreover, δ(·) is a Dirac delta function. The estimated value of the target parameter is expressed as:(17)$${\overset{⌢}{\Phi }}_{k}\approx \frac{\sum_{j=1}^{N_{p}}\omega_{k}^{\left(j\right)}P\left(x_{k}|\Phi_{k}^{\left(j\right)},{\bar{S}}_{k}\right)\Phi_{k}^{\left(j\right)}}{\sum_{i=1}^{N_{p}}\omega_{k}^{\left(j\right)}P\left(x_{k}|\Phi_{k}^{\left(j\right)},{\bar{S}}_{k}\right)}$$

The value of $P\left(x_{k}|\Phi_{k,i}^{\left(j\right)},{\bar{S}}_{k}\right)$ can be obtained by substituting the target parameter status $\Phi_{k,i}^{\left(j\right)}$ of hypothesis $H_{i}$ into Equation (13), and then the estimated value ${\overset{⌢}{\Phi }}_{k,i}$ can be obtained according to Equation (17).

According to Equation (14) for target parameter estimation, parameter ${\overset{⌢}{\Phi }}_{k}$ is related to the transmission waveform. Parameter ${\overset{⌢}{\Phi }}_{k}$ changes significantly make the decision condition $\Lambda_{i,j}^{\left(k\right)}$ of Equation (9) fluctuate, thus increasing the ASN. Then, efforts should be made to reduce the fluctuation of the estimated ${\overset{⌢}{\Phi }}_{k}$. Meanwhile, it is known if adaptive waveform design [8,11] is capable of reducing the estimation error of target parameters and reducing the ASN. However, the circumstance of low received SNR has not been considered earlier. From the perspective of the frequency domain [28], the fact that the waveform design under SNR constraint can improve the performance of the radar system was verified. However, the analysis was not performed within the time domain, and the focus was on the binary hypothesis test. In this study, SNR is considered as a constraint condition to design adaptive waveforms.

## 4. Waveform Design Based on Two-Stage Information Criterion

As analyzed, target parameter estimation and ASN reduction are key links in multi-hypothesis tests. Poor estimation accuracy can cause fluctuations in the decision criterion, thus increasing ASN and eventually degrading the target detection performance. Parameter estimation focuses on the target performance of a single hypothesis, whereas ASN is to solve the rapid decision between multi-hypotheses. Hence, the trade-off between the two can be solved in two stages.

### 4.1. Waveform Design for Parameter Estimation

As mentioned in Section 3, one of the key steps in multi-hypothesis tests is parameter estimation, and its performance can be improved by waveform design. Therefore, based on the constraint between the decision of multi-hypothesis tests and estimated parameters, waveform design is performed with MI as the criterion. According to the MIMO radar signal model, the MI between the received signal $x_{k}$ and the target response $h_{k}$ can be expressed as [18]:(18)$$\mathrm{MI}\left(x_{k};h_{k}\;|{\bar{S}}_{k}\right)=N_{R}\left\{\log\left[\det\left({\bar{S}}_{k}R_{k}{\bar{S}}_{k}^{H}+R_{N_{k}}\right)\right]-\log\left[\det\left(R_{N_{k}}\right)\right]\right\}$$

In multi-hypothesis tests, each hypothesis corresponds to $\mathrm{MI}\left(x_{k};h_{k}\;|{\bar{S}}_{k}\right)$, and waveform design can be conducted with MI maximization as a criterion. However, it is difficult to maximize the MI of all hypotheses by designing a waveform $\;{\bar{S}}_{k}$ based on a single hypothesis. Additionally, the probabilities of all hypotheses change because different observation signals are received in each pulse. The ultimate goal of waveform design is to concentrate hypothesis testing on true hypotheses and maximize the probability. If the waveform is optimized based on MI maximization, then a theoretical method to improve parameter estimation accuracy is to design a transmission waveform for multi-hypothesis tests based on the product sum of the posterior probability weights of all hypotheses and waveforms. This method is called a waveform based on the MI maximization of a single hypothesis.

According to [24], the mutual information is constrained by the SNR. When the target response covariance matrix $R_{k}$ is a fixed value, the energy constraint is equivalent to the SNR constraint. However, the waveform design needs to match the change in the target response, hence both the energy constraint and the SNR constraint need to be considered. In this study, the maximum transmission energy of the radar is set as $P_{0}$ and the SNR threshold as $\mathrm{SN}R_{0}$. Under the SNR and energy constraints, the objective function of waveforms designed based on the maximization of MI can be expressed as:(19)$$\begin{array}{l} \underset{{\bar{S}}_{k}}{\max\;}\mathrm{MI}=\log\left[\det\left(I_{K}+{\bar{S}}_{k}R_{k}{\bar{S}}_{k}^{H}R_{N_{k}}^{-1}\right)\right]\\ s.t.\;\;\mathrm{tr}\left({\bar{S}}_{k}R_{k}{\bar{S}}_{k}^{H}R_{N_{k}}^{-1}\right)\leq \mathrm{SN}R_{0}\\ \;\;\;\;\;\mathrm{tr}\left({\bar{S}}_{k}{\bar{S}}_{k}^{H}\right)\leq P_{0} \end{array}$$

Considering that the objective function (19) is a non-convex function, conversion processing is required. To see this, if we introduce an auxiliary variable F, it is easy to verify that the constraint ${\bar{S}}_{k}R_{k}{\bar{S}}_{k}^{H}R_{N_{k}}^{-1}\underline{≻}F\underline{≻}0$ is non-convex with respect to ${\bar{S}}_{k}$. The notation $A\underline{≻}B$ (A≻B) means A−B is positive definite (semi-definite). As a result, the optimization problem in (19) is non-convex. 

Let the singular value decomposition (SVD) of ${\bar{S}}_{k}$ be ${\bar{S}}_{k}=U_{{\bar{S}}_{k}}\Sigma_{{\bar{S}}_{k}}V_{{\bar{S}}_{k}}^{H}$, where $\Sigma_{{\bar{S}}_{k}}={\left[{\left(\Sigma_{S1}\right)}^{1/2},0_{N\times \left(N_{R}L-N\right)}\right]}^{T}$, $\Sigma_{S1}=\mathrm{Diag}\left({\left[\sigma_{s1},\cdots ,\sigma_{sN}\right]}^{T}\right)$, $\sigma_{s1}\geq \sigma_{s2}\geq \cdots \geq \sigma_{sN}$ and $N=N_{T}+N_{R}-1$. Here, ${\left(\sigma_{sk}\right)}^{1/2}$ is the *k*-th singular value of the waveform ${\bar{S}}_{k}$, and the *k*-th column vectors of $U_{{\bar{S}}_{k}}$ and $V_{{\bar{S}}_{k}}$ correspond to the singular vectors on the left and right sides of the eigenvalues ${\left(\sigma_{sk}\right)}^{1/2}$, respectively. Accordingly, the objective function (19) is converted based on the above decomposition, as shown below.
(20)$$\underset{V_{{\bar{S}}_{k}},\Sigma_{S1}}{\max}\mathrm{MI}=\log\left[\det\left(I_{K}+R_{k}V_{{\bar{S}}_{k}}\Sigma_{S1}V_{{\bar{S}}_{k}}^{H}\right)\right]$$

Similarly, the SNR constraint and energy constraints are simplified as:(21)$$\mathrm{tr}\left(R_{k}V_{{\bar{S}}_{k}}\Sigma_{S1}V_{{\bar{S}}_{k}}^{H}\right)\leq \mathrm{SN}R_{0}$$
(22)$$\mathrm{tr}\left({\bar{S}}_{k}{\bar{S}}_{k}^{H}\right)=\overset{N}{\underset{k=1}{\sum }}\sigma_{sk}\leq P_{0}$$

Then, the objective function (19) for waveform design is transformed into the following:(23)$$\begin{array}{l} \underset{V_{{\bar{S}}_{k}},\Sigma_{S1}}{\max}\;\log\left[\det\left(I_{K}+\Sigma_{S1}V_{{\bar{S}}_{k}}^{H}R_{k}V_{{\bar{S}}_{k}}\right)\right]\\ s.t.\;\;\mathrm{tr}\left(\Sigma_{S1}V_{{\bar{S}}_{k}}^{H}R_{k}V_{{\bar{S}}_{k}}\right)\leq \mathrm{SN}R_{0}\\ \;\;\;\;\;\overset{N}{\underset{k=1}{\sum }}\sigma_{sk}\leq P_{0} \end{array}$$

As shown by functions (19) and (23), the problem of waveform design is transformed from the optimization of ${\bar{S}}_{k}$ to that of $\Sigma_{S1}$ and $V_{{\bar{S}}_{k}}$. Perform an eigenvalue decomposition of $R_{k}=V_{R_{k}}\Sigma_{R_{k}}V_{R_{k}}^{H}$, where $\Sigma_{R_{k}}=\mathrm{Diag}\left({\left[\sigma_{H1},\cdots ,\sigma_{HN}\right]}^{T}\right)$ and $\sigma_{H1}\geq \sigma_{H2}\geq \cdots \geq \sigma_{HN}$. The *k*-th column vector of $V_{R_{k}}$ corresponds to the eigenvector of the eigenvalue $\sigma_{Hk}$. Accordingly, the objective function (23) is converted, as shown below.
(24)$$\;\log\left[\det\left(I_{K}+\Sigma_{S1}V_{{\bar{S}}_{k}}^{H}R_{k}V_{{\bar{S}}_{k}}\right)\right]=\log\left[\det\left(\Sigma_{S1}^{-1}+V_{{\bar{S}}_{k}}^{H}R_{k}V_{{\bar{S}}_{k}}\right)\right]+\log\left[\det\left(\Sigma_{S1}\right)\right]$$

It is obvious that $\Sigma_{S1}V_{{\bar{S}}_{k}}^{H}R_{k}V_{{\bar{S}}_{k}}$ and $I_{K}$ are both positive semidefinite Hermitian matrices. According to the inequality condition, $\overset{n}{\underset{i=1}{\prod }}\alpha_{i}+\beta_{i}\leq \det\left(A+B\right)\leq \overset{n}{\underset{i=1}{\prod }}\alpha_{i}+\beta_{n+1-i}$ of Lemma 1 in [18], we can ascertain that
(25)$$\;\log\left[\det\left(I_{K}+\Sigma_{S1}V_{{\bar{S}}_{k}}^{H}R_{k}V_{{\bar{S}}_{k}}\right)\right]\leq \overset{N}{\underset{k=1}{\sum }}\log\left(1+\sigma_{sk}\sigma_{Hk}\right)$$
where the equality is achieved if and only if $V_{R_{k}}=V_{{\bar{S}}_{k}}$. Then, ${\bar{S}}_{k}=U_{{\bar{S}}_{k}}\Sigma_{{\bar{S}}_{k}}V_{R_{k}}^{H}$. Similarly, the SNR constraint is simplified, as shown below.
(26)$$\mathrm{tr}\left(\Sigma_{S1}V_{{\bar{S}}_{k}}^{H}R_{k}V_{{\bar{S}}_{k}}\right)=\overset{N}{\underset{k=1}{\sum }}\sigma_{sk}\sigma_{Hk}\leq \mathrm{SN}R_{0}$$

Hence, the objective function (23) of waveform optimization is further converted into the following:(27)$$\begin{array}{l} \underset{\sigma_{sk}}{\max}\overset{N}{\underset{k=1}{\sum }}\log\left(1+\sigma_{sk}\sigma_{Hk}\right)\\ s.t.\;\overset{N}{\underset{k=1}{\sum }}\sigma_{sk}\sigma_{Hk}\leq \mathrm{SN}R_{0}\\ \;\;\;\;\;\overset{N}{\underset{k=1}{\sum }}\sigma_{sk}\leq P_{0} \end{array}$$

Notably, the objective function is a convex function where the constraints are linearly correlated with $\sigma_{sk}$. Thus, the optimization problem is a convex optimization, which can be solved by the method of Lagrange multipliers, i.e.,
(28)$$L\left(\sigma_{sk},\lambda_{1},\lambda_{2}\right)=-\overset{N}{\underset{k=1}{\sum }}\log\left(1+\sigma_{sk}\sigma_{Hk}\right)+\lambda_{1}\left(\overset{N}{\underset{k=1}{\sum }}\sigma_{sk}\sigma_{Hk}-\mathrm{SN}R_{0}\right)+\lambda_{2}\left(\overset{N}{\underset{k=1}{\sum }}\sigma_{sk}-P_{0}\right)$$
where $\lambda_{1}\geq 0$ and $\lambda_{2}\geq 0$ correspond to the Lagrange multipliers of the SNR constraint and energy constraints, respectively. The derivative of $L\left(\sigma_{sk},\lambda_{1},\lambda_{2}\right)$ with respect to $\sigma_{sk}$ is solved, and the derivative is zero. Then the solution to the equation of $\sigma_{sk}$ can be expressed as:(29)$$\sigma_{sk}={\left[\frac{1}{\lambda_{1}\sigma_{Hk}+\lambda_{2}}-\frac{1}{\sigma_{Hk}}\right]}^{+}$$

From the expression of ${\bar{S}}_{k}=U_{{\bar{S}}_{k}}\Sigma_{{\bar{S}}_{k}}V_{R_{k}}^{H}$, for the optimized waveform obtained based on the MI maximization criterion, the right singular vector of the matrix generally matches the eigenvector of the target response matrix. Then, the optimized waveform is expressed as:(30)$${\bar{S}}_{k}=U_{{\bar{S}}_{k}}{\left\{\mathrm{diag}{\left[{\left(\frac{1}{\lambda_{1}\sigma_{H1}+\lambda_{2}}-\frac{1}{\sigma_{H1}}\right)}^{+},\ldots ,{\left(\frac{1}{\lambda_{1}\sigma_{HN}+\lambda_{2}}-\frac{1}{\sigma_{HN}}\right)}^{+}\right]}^{1/2},0_{N\times \left(N_{R}L-N\right)}\right\}}^{T}V_{R_{k}}^{H}$$

Without the SNR constraint, the optimized waveform can be expressed as:(31)$${\bar{S}}_{k}=U_{{\bar{S}}_{k}}{\left\{\mathrm{diag}{\left[{\left(\frac{1}{\lambda_{1}}-\frac{1}{\sigma_{H1}}\right)}^{+},\ldots ,{\left(\frac{1}{\lambda_{1}}-\frac{1}{\sigma_{HN}}\right)}^{+}\right]}^{1/2},0_{N\times \left(N_{R}L-N\right)}\right\}}^{T}V_{R_{k}}^{H}$$

This result is in accordance with that in [21]. The optimized waveform of each hypothesis is obtained, and the posterior probability $P_{i}^{k}\left(H_{i}|x_{k}\right)$ of each hypothesis during the *k*-th waveform transmission is generated based on the received echo signal. Then, the (*k + 1*)-th transmission waveform based on the single hypothesis of MI maximization is:(32)$${\bar{S}}_{k+1}^{\mathrm{SGMI}}=\overset{M-1}{\underset{i=0}{\sum }}P_{i}^{k}\left(H_{i}|x_{k}\right){\bar{S}}_{ik}$$
where M represents the total number of hypotheses, and ${\bar{S}}_{ik}$ indicates the optimized waveform of the *i*-th hypothesis $H_{i}$. The posterior probability $P_{i}^{k}\left(H_{i}|x_{k}\right)$ of each hypothesis experiences an iterative update according to the previous (*k* − 1)-th transmission waveform and the received echo. Hence, the posterior probability $P_{i}^{k}\left(H_{i}|x_{k}\right)$ of hypothesis $H_{i}$ can be expressed as:(33)$$P_{i}^{k}\left(H_{i}|x_{k}\right)=\frac{P_{i}^{k-1}\left(x_{1},x_{2},\cdots ,x_{k-1}\right)P_{i}^{\left(0\right)}}{\sum_{i=0}^{M-1}P_{i}^{k-1}\left(x_{1},x_{2},\cdots ,x_{k-1}\right)P_{i}^{\left(0\right)}}$$

Here, $P_{i}^{\left(0\right)}$ is the prior probability of hypothesis $H_{i}$, and $P_{i}^{k-1}\left(x_{1},x_{2},\cdots ,x_{k-1}\right)$ denotes the joint probability density function of the (*k* − 1)-th transmission waveform, which can be further expressed as the product of the probability density functions with target parameters and transmission waveforms as the conditions, as shown below.
(34)$$P_{i}^{k-1}\left(x_{1},x_{2},\cdots ,x_{k-1}\right)=\overset{k-1}{\underset{t=1}{\prod }}P_{i}\left(x_{t}|\Phi_{t,i}\right)$$

### 4.2. Waveform Design of ASN Reduction

In multi-hypothesis tests, ASN is a key factor for target detection, and it fluctuates with the accuracy of target parameter estimation. The estimated target parameters directly affected the posterior PDF $P_{i}^{k}\left(H_{i}|x_{k}\right)$ through the $P_{i}\left(x_{k}|\Phi_{k,i}\right)$ of each hypothesis. ASN reduction can be performed by two approaches. One is to minimize the MI between the posterior probabilities $P_{i}^{k}\left(H_{i}|x_{k}\right)$ of the multi-hypotheses. For simple binary hypotheses testing, maximizing the KLD between two distributions of two hypotheses can obtain optimal detection performance, according to Stein’s lemma [31,32]. Thus, the other is to maximize the KLD between the posterior probabilities $P_{i}^{k}\left(H_{i}|x_{k}\right)$ of the multi-hypotheses to increase the distance metric between the hypotheses. As ASN reduction occurs after parameter estimation, during waveform optimization, the waveform designed based on the criterion of MI maximization in the first stage should be used as the library for waveform selection. On this basis, a multi-target detection waveform for ASN reduction is designed.

#### 4.2.1. Waveform Design Based on Dual-Hypothesis MI Minimization

The amount of information between the posterior probabilities of each hypothesis can be reduced by minimizing the MI between the received echoes through multiple hypotheses. Thus, according to the definition of MI, the MI between $x_{ik}$ and $x_{jk}$ (the output signal of hypothesis $H_{i}$ and hypothesis $H_{j}$, respectively) can be expressed as:(35)$$\mathrm{MI}\left(x_{ik},x_{jk}\right)=H\left(x_{ik}|{\bar{S}}_{k}\right)+H\left(x_{jk}|{\bar{S}}_{k}\right)-H\left(x_{ik},x_{jk}|{\bar{S}}_{k}\right)$$

Combining the entropy calculation formula, the following equation is obtained [19]:(36)$$\mathrm{MI}\left(x_{ik},x_{jk}\right)=-N\ln\left[\det\left(I_{N}-D^{2}\right)\right]$$
where $I_{N}$ refers to the identity matrix of order N, and D is the diagonal matrix after SVD of the covariance matrix $R_{H_{i},H_{j}}$, as shown below.
(37)$$R_{H_{i},H_{j}}={\left(\sqrt{R_{x_{ik}}^{-1}}\right)}^{H}R_{x_{ik},x_{jk}}\sqrt{R_{x_{jk}}^{-1}}$$
where $R_{x_{ik}}$, $R_{x_{jk}}$ and $R_{x_{ik},x_{jk}}$ are calculated as:(38)$$R_{x_{ik}}={\bar{S}}_{k}R_{ik}{\bar{S}}_{k}^{H}+R_{N_{k}}$$
(39)$$R_{x_{jk}}={\bar{S}}_{k}R_{jk}{\bar{S}}_{k}^{H}+R_{N_{k}}$$
(40)$$R_{x_{ik},x_{jk}}={\bar{S}}_{k}h_{k}{\left(\Phi_{ik}\right)}^{H}h_{k}\left(\Phi_{jk}\right){\bar{S}}_{k}^{H}$$

The waveform optimization criterion can be expressed as:(41)$$\begin{array}{l} \underset{{\bar{S}}_{ijk}}{\min}\;\mathrm{MI}\left(x_{ik},x_{jk}\right)=-N\ln\left[\det\left(I_{N}-D^{2}\right)\right]\\ \;\;s.t.{\bar{S}}_{ijk}\in \left\{{\bar{S}}_{ik},{\bar{S}}_{jk}\right\} \end{array}$$

Given that the optimized waveform is mainly for the MI between hypothesis $H_{i}$ and hypothesis $H_{j}$, only the waveform with the best parameter estimation performance is selected for the construction of the waveform library, i.e., $S=\left\{{\bar{S}}_{ik},{\bar{S}}_{jk}\right\}$. Consequently, the waveform designed based on dual-hypothesis MI minimization is expressed as:(42)$${\bar{S}}_{k+1}^{\mathrm{DBMI}}=\overset{M-1}{\underset{i=0}{\sum }}\;\overset{M-1}{\underset{j=0}{\sum }}P_{i}^{k}\left(H_{i}|x_{k}\right)P_{j}^{k}\left(H_{j}|x_{k}\right){\bar{S}}_{ijk}$$

#### 4.2.2. Waveform Design Based on Dual-Hypothesis KLD Maximization

In multi-hypothesis tests, ASN reduction is also possible by increasing the relative entropy between hypotheses. In [17], adaptive waveforms were designed through the maximization of the KLD between all hypotheses, where the objective function was the product sum of the weight of the assumed probability density function and the J-divergence between any two hypotheses, i.e., $\underset{{\bar{S}}_{k}}{\max}\sum_{i=0}^{M-1}\sum_{j=i+1}^{M-1}P_{i}^{k}\left(H_{i}|x_{k}\right)P_{j}^{k}\left(H_{j}|x_{k}\right)d\left(H_{i};H_{j}\right)$. The J-divergence $d\left(H_{i};H_{j}\right)$ between any two hypotheses is expressed as the KLD sum of the posterior probability, i.e., $d\left(H_{i};H_{j}\right)=D\left(P_{i}^{k}||P_{j}^{k}\right)+D\left(P_{j}^{k}||P_{i}^{k}\right)$. In their method, the KLD between all hypotheses need to be calculated, and the calculation amount is large in the case of the rising hypotheses. According to [33], the ASN of each hypothesis is inversely proportional to $\underset{j:j\neq i}{\min}D\left(P_{i}^{k}||P_{j}^{k}\right)$. Therefore, the objective function of waveform optimization based on dual-hypothesis KLD maximization can be expressed as:(43)$$\underset{{\bar{S}}_{k}}{\max}\sum_{i=0}^{M-1}\sum_{j=i+1}^{M-1}P_{i}^{k}\left(H_{i}|x_{k}\right)P_{j}^{k}\left(H_{j}|x_{k}\right)\underset{j:j\neq i}{\min}D\left(P_{i}^{k}||P_{j}^{k}\right)$$

According to the definition of relative entropy, the KLD between any two hypotheses is:(44)$$\begin{array}{cc} D\left(P_{i}^{k}||P_{j}^{k}\right) & =\log\det\left({\bar{S}}_{k}R_{jk}{\bar{S}}_{k}^{H}+R_{N_{k}}\right)-\log\det\left({\bar{S}}_{k}R_{ik}{\bar{S}}_{k}^{H}+R_{N_{k}}\right)+\mathrm{tr}\left[{\left({\bar{S}}_{k}R_{jk}{\bar{S}}_{k}^{H}+R_{N_{k}}\right)}^{-1}\left({\bar{S}}_{k}R_{ik}{\bar{S}}_{k}^{H}+R_{N_{k}}\right)\right]\\ \; & +{\left({\bar{S}}_{k}h_{ik}-{\bar{S}}_{k}h_{jk}\right)}^{H}{\left({\bar{S}}_{k}R_{jk}{\bar{S}}_{k}^{H}+R_{N_{k}}\right)}^{-1}\left({\bar{S}}_{k}h_{ik}-{\bar{S}}_{k}h_{jk}\right) \end{array}$$

In the event of multiple hypotheses, the mean value of the received echo under each hypothesis is ${\bar{S}}_{k}h_{ik}$, and unlike the expression of KLD in [25], the expression ${\left({\bar{S}}_{k}h_{ik}-{\bar{S}}_{k}h_{jk}\right)}^{H}{\left({\bar{S}}_{k}R_{jk}{\bar{S}}_{k}^{H}+R_{N_{k}}\right)}^{-1}\left({\bar{S}}_{k}h_{ik}-{\bar{S}}_{k}h_{jk}\right)$ is introduced. According to the properties of matrix transformation, the additional expression is converted as:(45)$${\left({\bar{S}}_{k}h_{ik}-{\bar{S}}_{k}h_{jk}\right)}^{H}{\left({\bar{S}}_{k}R_{jk}{\bar{S}}_{k}^{H}+R_{N_{k}}\right)}^{-1}\left({\bar{S}}_{k}h_{ik}-{\bar{S}}_{k}h_{jk}\right)=\mathrm{tr}\left[{\left({\bar{S}}_{k}R_{jk}{\bar{S}}_{k}^{H}+R_{N_{k}}\right)}^{-1}\left({\bar{S}}_{k}h_{ik}-{\bar{S}}_{k}h_{jk}\right){\left({\bar{S}}_{k}h_{ik}-{\bar{S}}_{k}h_{jk}\right)}^{H}\right]$$

Then, $D\left(P_{i}^{k}||P_{j}^{k}\right)$ can be expressed as:(46)$$D\left(P_{i}^{k}||P_{j}^{k}\right)=\log\det\left({\bar{S}}_{k}R_{jk}{\bar{S}}_{k}^{H}+R_{N_{k}}\right)-\log\det\left({\bar{S}}_{k}R_{ik}{\bar{S}}_{k}^{H}+R_{N_{k}}\right)+\mathrm{tr}\left[{\left({\bar{S}}_{k}R_{jk}{\bar{S}}_{k}^{H}+R_{N_{k}}\right)}^{-1}\left({\bar{S}}_{k}R_{Bk}{\bar{S}}_{k}^{H}+R_{N_{k}}\right)\right]$$
where $R_{ijk}=h_{ik}h_{jk}^{H}$ and
(47)$$R_{Bk}=2R_{ik}+R_{jk}-R_{ijk}-R_{ijk}^{H}$$

According to [25], when $M_{k}$ is a positive definite matrix, $\log\det\left(I+M_{k}\right)+\mathrm{tr}\left[{\left(I+M_{k}\right)}^{-1}\right]$ increases monotonically with $M_{k}$. The covariance matrix of the target parameters $R_{jk}$ is a Hermitian matrix involving the attenuation amplitude and the target angle. According to Equation (46) and the theorem of matrix positive definiteness, $D\left(P_{i}^{k}||P_{j}^{k}\right)$ increases monotonically with $h_{jk}$ or $R_{jk}$ when $R_{jk}\geq R_{ik}$ (i.e., $h_{jk}\geq h_{ik}$); otherwise, it decreases monotonically with $h_{jk}$ or $R_{jk}$ when $R_{jk}<R_{ik}$. Therefore, only the value of *j* at $R_{jk}\geq R_{ik}$ should be considered to determine the $\underset{j:j\neq i}{\min}D\left(P_{i}^{k}||P_{j}^{k}\right)$ between hypothesis $H_{i}$ and hypothesis $H_{j}$. The covariance matrix of the target parameters for each hypothesis is determined by the direction angle $\theta_{k}$ and attenuation amplitude $\alpha_{k}$ of the target, where all elements are related to ${\left(h_{k}\right)}^{i}$. Hence, such a matrix can be sorted in decreasing order. If $\cdots \geq R_{tk}\geq R_{ik}\geq R_{ck}\geq \cdots$, $\underset{j:j\neq i}{\min}D\left(P_{i}^{k}||P_{j}^{k}\right)$ indicates the KLD between the *i*-th and the *t*-th hypotheses, as shown below.
(48)$$\underset{j:j\neq i}{\min}D\left(P_{i}^{k}||P_{j}^{k}\right)=D\left(P_{i}^{k}||P_{t}^{k}\right),t\in 0,1,\cdots M-1,t\neq i$$

Then, the objective function (43) can be transformed into the following:(49)$$\underset{{\bar{S}}_{k}}{\max}\sum_{i=0}^{M-1}P_{i}^{k}\left(H_{i}|x_{k}\right)P_{t}^{k}\left(H_{t}|x_{k}\right)D\left(P_{i}^{k}||P_{t}^{k}\right),t\in 0,1,\cdots M-1,t\neq i$$

The core of the waveform optimization objective function (49) is to maximize $D\left(P_{i}^{k}||P_{t}^{k}\right)$. In the event of $\underset{{\bar{S}}_{itk}\in S}{\max}D\left(P_{i}^{k}||P_{t}^{k}\right),t\in 0,1,\cdots M-1,t\neq i$, the obtained waveform is ${\bar{S}}_{itk}$, the ASN of hypothesis $H_{i}$ is the smallest, and the waveform library is $S=\left\{{\bar{S}}_{ik},{\bar{S}}_{tk}\right\}$. Then, the waveform design based on dual-hypothesis KLD maximization can be expressed as:(50)$${\bar{S}}_{k+1}^{\mathrm{DBKLD}}=\sum_{i=0}^{M-1}P_{i}^{k}\left(H_{i}|x_{k}\right)P_{t}^{k}\left(H_{t}|x_{k}\right){\bar{S}}_{itk},t\in 0,1,\cdots M-1,t\neq i$$

As analyzed above, the algorithm of waveform optimization based on the two-stage information criterion is described in Table 1.

## 5. Simulation and Results

In this section, the parameter estimation accuracy and the detection probability are mainly used as the indicators of multi-hypothesis tests to verify the waveform design based on the two-stage information criterion. In the first stage, a waveform with SNR constraint is designed based on single-hypothesis MI maximization. The estimate waveform without SNR constraint [18] is selected for contrast in parameter estimation. In the second stage, waveform selection is performed based on the estimated waveform library of the first stage. The waveform is designed with KLD maximization as the criterion [17], which is used as a comparison scheme for reduced ASN.

When the waveform with SNR constraint designed in the first stage is used as the waveform library, it is in contrast to the transmission waveform designed based on single-hypothesis MI maximization (SGMI waveform), dual-hypothesis MI minimization (DBMI waveform), dual-hypothesis KLD maximization (DBKLD waveform), and the KLD criterion [17] (DBKLDRef waveform). When the waveform without SNR constraint in the first stage is used as the waveform library, it is in contrast to the transmission waveform designed based on single-hypothesis MI maximization (SGMI-NoSNR waveform), dual-hypothesis MI minimization (DBMI-NoSNR waveform), dual-hypothesis KLD maximization (DBKLD-NoSNR waveform), and the KLD criterion [17] (DBKLDRef-NoSNR waveform). Furthermore, the orthogonal waveform is added as a benchmark to verify the waveform designed in this paper.

In this simulation, the target of MIMO radar detection is treated as point scatter, and the receive and transmit arrays are uniform and linear with $N_{T}=N_{R}=6$ elements, with half wavelength inter-element spacing for both transmit and receive arrays. We consider a scenario where the MIMO radar must estimate the angles of three targets. The attenuation amplitude of waveform signals remains the same after being reflected by three assumed targets in space, which is 1 dB, but the angles of the three targets are different. The deviation angles of the three targets from the beam center line are assumed to be −33°, 0°, and 30°, respectively, and the SNR constraint threshold $\mathrm{SN}R_{0}$ is set as −3 dB.

### 5.1. Performance of Parameter Estimation

As the design of adaptive waveforms is the focus of this study, rather than the verification of the estimation performance of the filter, the particle filter is simplified, and we initialize the particles on a grid equally spaced, i.e., $N_{p}=180$. In the initialization stage, equal weight allocation is conducted on spatial angles between −90° and 89°, and each particle is assigned a weight of 1/180. With changes in the adaptive waveform, the particles $\omega_{k}^{\left(i\right)}$ are resampled based on a Gaussian distribution, where the mean of the received data is considered as the mean value, and the reduction factor $t^{k-1}\;,\;t<1$ is the variance, where t=0.75. Then the parameters are estimated according to the designed waveform. Figure 1 presents the comparison result of the MMSE obtained by target parameter estimation.

As observed in Figure 1, the MMSE of the target parameters decreases with increasing pulse number. Furthermore, because the first stage is parameter estimation based on the MI criterion and the second stage is waveform selection from the library of parameter estimation, in general, the MMSE of target parameters is reduced under the influence of the waveform designed based on the MI maximization criterion.

In Figure 1, the waveform without SNR constraint outperforms that with SNR constraint in parameter estimation, whereas the orthogonal waveform exhibits inferior performance than the waveforms in the other two cases. The SGMI waveform with SNR constraint performs better than that without SNR constraint, which was mainly due to its larger fluctuation range of each hypothesis probability. The SNR constraint reduces the fluctuations caused by the product sum of the hypothesis probability and the waveform. The DBMI-NoSNR waveform achieves the best performance because MI is used as the optimization criterion in both stages, which focuses on target information extraction.

At the initial moment of the simulation, the MMSE of the target parameter is 30°, and under the multi-hypothesis probability iterations, the MMSE of the target parameter is reduced at a high rate. The DBKLD-NoSNR waveform exhibits better performance before the 11th pulse, with a high decline rate, which results from the fact that the KLD criterion rapidly separates the distance between probabilities. Nonetheless, from the perspective of parameter estimation, the KLD criterion is inferior to the MI criterion, and this finding is in accordance with [28]. At the 25th pulse, the target parameter MMSE of the orthogonal waveform reaches 2°, while the target parameter MMSE of the DBMI-NoSNR waveform can reach 0.35°. The DBMI-NoSNR and DBKLD-NoSNR waveforms proposed in this paper outperform the DBKLDRef-NoSNR waveform [17].

### 5.2. Performance of Target Detection

Although three targets in the space are illuminated by the radar waveform, seven hypotheses contain targets in multi-hypothesis tests. Among them, hypotheses $H_{1}$, $H_{2}$, and $H_{3}$ contain one target; hypotheses $H_{4}$, $H_{5}$, and $H_{6}$ contain two; and hypothesis $H_{7}$ contains three, with the error probability of $\varepsilon_{ij}=0.01$. During the sequential detection, the initial probability of each hypothesis is $P_{i}^{\left(0\right)}=1/7$.

The initial transmission waveform is designed without prior information, and the parameter $\Phi_{k}$ is estimated based on the first echo signal. As the waveform is adaptively adjusted, the probability of each hypothesis is updated with change in the output data. Through the simulation, the DBMI-NoSNR waveform achieves a correct decision hypothesis $H_{7}$ when transmitting the 25th pulse, as shown in Figure 2.

As observed in Figure 2, the probability of each hypothesis fluctuates with increasing pulse number, which is related to the change in the target response covariance matrix in the hypothesis. The estimated parameters change after the transmission of each waveform, thus leading to fluctuations in the probability of each hypothesis. The relations of the DBMI-NoSNR waveform and the corresponding angular PDF with the transmitted pulse number are depicted in Figure 3.

According to Figure 3, with an increase in the pulse number, the targets present a more concentrated probability distribution at three angles: −33°, 0°, and 30°, reaching the maximum probability distribution at pulse number 25. However, when the pulse number is 14, the three targets already show a concentrated angular probability distribution. In this case, the probability of $H_{7}$ is higher than that of the other hypotheses, approaching 1. Before the 14th pulse, the mean square error of the target parameters changes more sharply with faster improvement in detection performance, suggesting that adaptive waveform design always supersedes changes in the target detection performance and estimation performance. The target detection performance of the nine adaptive waveform types is presented in Figure 4.

Figure 4 depicts that as the parameter estimation accuracy is improved with an increase in the pulse number, the SNR obtained by the current pulse and the detection probability both increase simultaneously, and the DBKLD-NoSNR waveform achieves the best target detection performance. This is because this waveform enables the estimated parameters to meet the corresponding requirements faster. However, at a small pulse number, its detection performance is weaker than that of the DBMI-NoSNR waveform. The comparison results of the detection performance are in accordance with the conclusion in [28], i.e., the KLD waveform is more suitable for improving the detection performance. As the detection probability is significantly influenced by the target parameter estimation, the detection probability of the designed waveform in this paper is directly proportional to the parameter estimation performance.

## 6. Conclusions

To improve multi-target detection performance, the constraint relations of the hypothesis test decision with parameter estimation and ASN are analyzed in this study. Next, a waveform design based on single-hypothesis MI maximization under an SNR constraint is proposed to enhance the parameter estimation performance. On this basis, a waveform design method based on the criterion of dual-hypothesis MI minimization and KLD maximization is proposed to reduce the ASN, thus realizing a two-stage waveform design based on the information theoretic criteria. The simulation results show that the waveform design based on the MI minimization criterion performs better in parameter estimation throughout the waveform design. Due to the small error probability in the sequential test, the decision frequency fails to match the target detection probability. Therefore, a direction for future waveform design lies in exploring a method facilitating adaptive changes in the decision threshold of hypothesis tests.

## Figures and Tables

**Figure 1 entropy-24-01075-f001:**
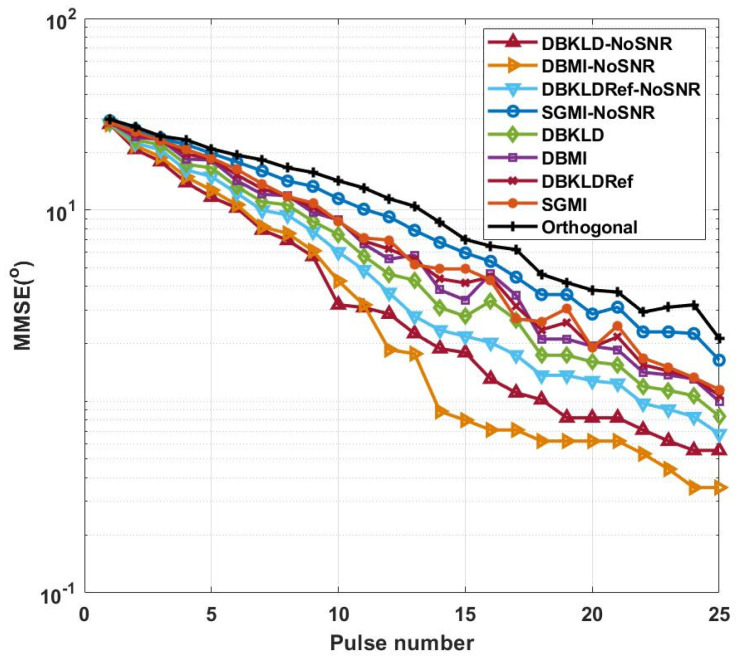
Pulse number-dependent changes in the target parameter MMSE.

**Figure 2 entropy-24-01075-f002:**
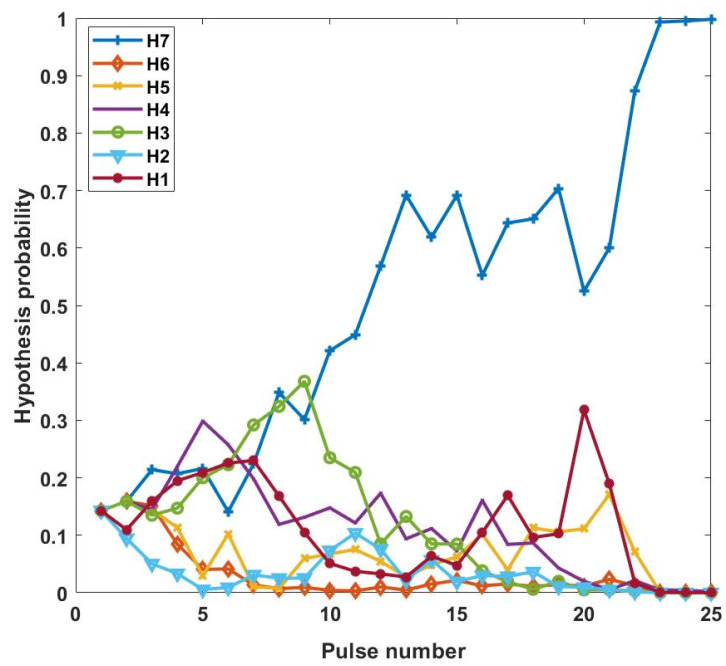
Pulse number-dependent changes in hypothesis probability.

**Figure 3 entropy-24-01075-f003:**
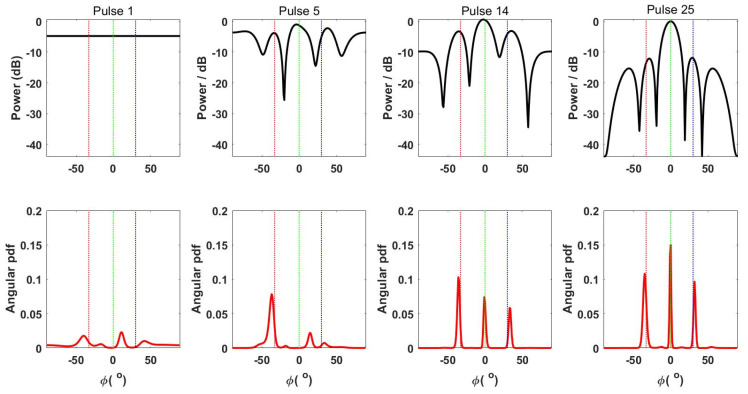
Pulse number-dependent changes in the DBMI-NoSNR waveform and angular PDF.

**Figure 4 entropy-24-01075-f004:**
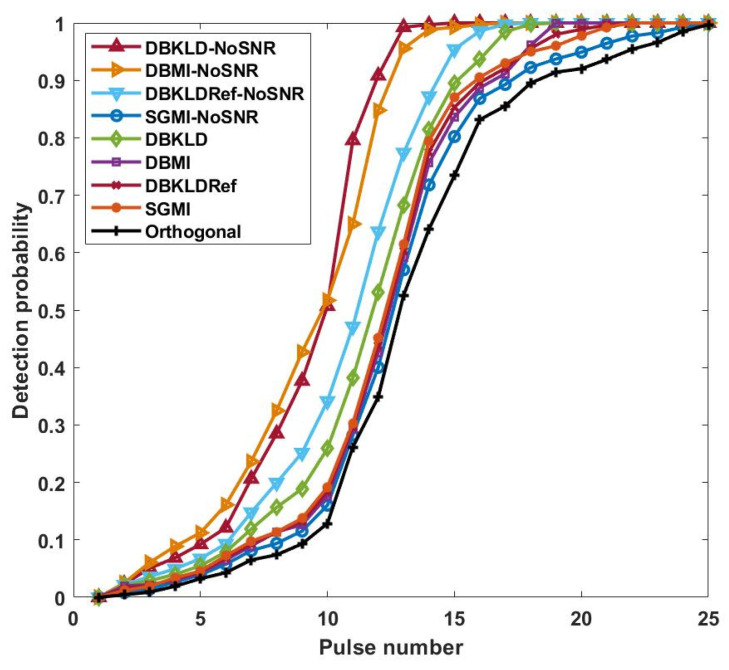
Pulse number-dependent change in target detection probability.

**Table 1 entropy-24-01075-t001:** Waveform optimization based on two-stage information criterion.

Input: $R_{N_{k}}$, $P_{0}$, $\mathrm{SN}R_{0}$, $\varepsilon_{ij}$, ${\bar{S}}_{0}$, $\theta_{0}$, $\alpha_{0}$ Initialize: k=0, the prior probability of each hypothesis is $P_{i}^{\left(0\right)}$. Design a hypothesis testing model according to Equation (7).
For: k=1:K
Calculate $x_{k}$ for each hypothesis based on Equation (6). Estimate parameter $\Phi_{k}$ for each hypothesis based on Equation (17) and solve $R_{ik}$ based on Equation (5).
Calculate $P_{i}\left(x_{k}|\Phi_{k,i}\right)$ for each hypothesis based on Equation (13). Calculate $\Lambda_{i,j}^{\left(k\right)}$ based on Equation (9). If Equation (11) is satisfied, the hypothesis is judged to be $H_{i}$, end loop; otherwise, continue the calculation process.
Update the transmission waveform ${\bar{S}}_{k}$ of each hypothesis based on Equation (30). Update the posterior probability $P_{i}^{k}\left(H_{i}|x_{k}\right)$ of each hypothesis based on Equation (33). Design the waveform in the following three cases: 1. Design the transmission waveform ${\bar{S}}_{k+1}^{\mathrm{SGMI}}$ based on single-hypothesis MI maximization as shown in object function (32). 2. Design the transmission waveform ${\bar{S}}_{k+1}^{\mathrm{DBMI}}$ based on dual-hypothesis MI minimization as shown in object function (42).
3. Design the transmission waveform ${\bar{S}}_{k+1}^{\mathrm{DBKLD}}$ based on dual-hypothesis KLD maximization as shown in object function (50).

## Data Availability

Not applicable.

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
