# Peer review of "Waveform Design for Multi-Target Detection Based on Two-Stage Information Criterion"

_entropy, 2022, doi:10.3390/e24081075_

Round 1

Reviewer 1 Report

This is a well-written paper that should be published.  I only have a few minor comments and suggestions as follows:

1.  On lines 71-72, the sentence “considering that the MI…” is not a complete sentence.

2.  On lines 89 and 105, you use the term “information theoretic”. At least to me, this term seems incomplete.  Should not an additional word be used such as “measure” or “criterion”?

3.  On line 146, the letter j needs to be defined.

4.  A brief additional explanation or justification for equation (9) would be helpful.

5.  I suggest that you present equation (11) immediately before equation (10) (and renumber them).

6.  On line 294, you state that “the objective function (19) is a non-convex function”.  A brief explanation would be helpful.

7.  Paragraph indentations are used in a number of places throughout the paper where they are not needed or are inappropriate.  See, for example, lines 246, 303, 306, 340.

8.  On line 319, “Where” should be lower case.

9.  On line 325, you state that the sigma is “the optimization variable”.  I think it would be helpful to the readers if you could elaborate a bit on this.

10.  When you present equation (33), the P(with subscript i and superscript 0) ought to be defined again.

11.  While there are any number of measures of statistical “distance”, you are using the Kullback-Leibler divergence.  A brief explanation of why you chose KLD would be helpful.

12.  On line 420, 421, 422, and 426, the symbol should be greater or than equal to (you use a symbol that resembles the Schur-convexity (concavity) symbol).  Similarly, on line 421, the symbol should be < not your wavy symbol.

13.  In the “Simulation and results” section you present the specifications N=6 (on line 464), the deviation angles (on lines 466-467), and the number of particles set at 180 (line 472).  A brief explanation or justification for those choices would be desirable.

Author Response

Dear reviewer,

Thank you very much for giving us an opportunity to revise the manuscript entitled “Waveform Design for Multi-Target Detection Based on Two-Stage Information Criterion” (ID: entropy-1826647). We express our great appreciation to you for the constructive comments and suggestions.

We have responded to all comments and made corresponding modifications on the manuscript. Revised parts are marked in red in the revised manuscript. The main corrections and the responses to the comments are appended below.

Thank you very much for your work to our manuscript.

Yours sincerely,

Yu Xiao, XiaoXiang Hu

Reviewer 2 Report

The paper clearly apporaches wavefront design for target detection. It is well-writen, thus I would detail or reference some sections and explanations in order to facilitate the reading of the document. Some recommendations are as follows:

Check the formatting of equations and expressions in general.

Several equations like (1), (2), etc. whose deduction is not provided should be adequately referenced.

In general, the methods applied should be defended more concretely, via references, or detailed explanations. For pointing out some examples, when defining the parameters of the wavefront, I don't find that trivial the change from eq. (13) to (17)

In eq (19) is stated that the function is not convex, so a SVD is performed. How can you assure that the resulting transformation is convex and represents the original function adequately? Moreover, considering that when reachin eq(23) a further transformation with eigenvalue decomposition is required.

Results should be more detailed; Figure 1 shows the MMSE of the different situations consided, however, how big that MMSE is? Is there any MMSE value to be taken as comparison? which percentage of error is associated to that value? (which I supposse that is connected with the figure 3 probability of detection)

Author Response

(The authors gave the same response as above.)
